# Scientometric Analysis of Research on Corporate Social Responsibility

**Mario Morales-Parragué [1,2,\*], Luis Araya-Castillo [3] , Fidel Molina-Luque [4] and Hugo Moraga-Flores [5]**

1   Facultad de Economía, Gobierno y Comunicaciones, Universidad Central de Chile, Santiago 8320000, Chile
2   Law and Business Administration Program, University of Lleida, 25003 Lleida, Spain
3   Facultad de Economía y Negocios, Universidad Andrés Bello, Santiago 7591538, Chile; luis.araya@unab.cl
4   Facultad de Educación, Psicología y Trabajo Social (GESEC-INDEST), Universidad de Lleida, 25003 Lleida, Spain; fidel.molinaluque@udl.cat
5   Departamento de Contabilidad y Auditoría, Universidad de Concepción, Concepción 4070386, Chile; hmoraga@udec.cl
\*   Correspondence: mmorales@fen.uchile.cl; Tel.: +56-9-97428221

**Abstract:** This work shows how Corporative Social Responsibility (CSR) has been filtering into different management areas, providing an insight into its evolution, and presenting literature reviews and efforts to incorporate conceptualisations and recommendations on its application. It can be understood through a scientometric and bibliometric analysis, using the WoS documents on the "Social Responsibility" concept in the "Business and Economics" category, analysing a total of 8728 papers up to the year 2020. In this work, CSR is associated with views from different fields of study in economics and business, highlighting diverse management fields; it seeks to explain the correlation between CSR and concepts from such fields of study, suggesting that there is a need to order and question the current understanding of CSR and show its relevance so it can be considered an area of specialisation within the management of businesses.

**Keywords:** corporate social responsibility; CSR; stakeholders; management; scientometric analysis; VOSviewer

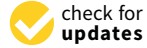



## 1. Introduction

Papers by Bowen [1] and Levitt [2] on Corporate Social Responsibility (CSR) sparked an academic, political, and business debate on its nature and implications for organisational management. From then on, efforts have been made to offer views and classifications that could help its understanding and contribute to the construction and application of the CSR concept [3–8], which seems to be capturing the interest of various management areas to be studied. The huge amount of scholarly output, the many efforts to categorise it, and the discussion to guide its understanding and study, pose a major academic challenge. The analysis of the field of study, i.e., CSR-related concepts, is an important step in planning research on this topic.

This work shows how CSR has been developing since its early academic debates, exploring current approaches, and studying CSR's relationship with other fields. The research starts by extensively analysing the literature that describes the efforts made by scholars over time in order to be able to understand and categorize CSR. It also shows how the literature is becoming increasingly complex as it relates to multiple areas of management. A bibliometric and scientometric analysis is then carried out to explore the growth of scientific production over time and contribute to the understanding of its current state. With these two combined studies, the idea is to question whether or not CSR should be considered as an area of specialisation within management and business, or as a fundamental alternative to organisational management and start a debate to build a common understanding of CSR and explore a new ways to apply it.

One of the main contributions to the construction of the CSR concept was made in 1970 when business social responsibility was discussed from a shareholder approach [4]. The definition of stakeholders [9] was then presented, extending the shareholder-centred responsibility to groups that affect or may affect the company.

A first attempt to classify CSR actions was conducted by Carroll [3] by dividing them into four types: Economic, Legal, Ethical, and Philanthropical, the last being the one where the aim is to contribute to society beyond the mere business side. Lantos [7] proposed for points of engagement in CSR activities: profit generation as a sole goal, profit generation in a limited scope, social welfare (where ethics come into play), and business in the service of the community. Garriga and Melé [5] studied the literature up to that year and proposed four groups of theories of CSR implementation: Instrumental theories, where CSR activities make sense if there is direct benefit to shareholders. Integrative theories, where CSR activities integrate social demands. Political theories, which emphasise the social power of corporations in society as they influence its economy. Ethical theories, with the prevalence of actions for the common good and the fulfilment of universal rights.

Later Porter and Kramer [8] proposed the concept of shared value as a new way of relating the company to its environment and from there apply CSR. Additionally, in 2015, the United Nations Global Compact outlined 17 Sustainable Development Goals (SDGs), wanting governments and private companies to cooperate to achieve goals associated with critical results for the management of CSR practices [10]. Based on the SDGs, Landrum and Ohsowski [6] included CSR ideas in terms of Corporate Sustainability and the message that companies convey through these actions. They classified companies into five sequential and evolutionary stages of CSR implementation: Compliance (very weak sustainability), where the company must participate in activities promoted by external actors; Business-Centred (weak sustainability), where the company participates in internal activities for its own benefit; Systemic (intermediate sustainability), where the company works with other actors integrating all elements of sustainability activities; Regenerative (strong sustainability), where the company understands sustainability and seeks to repair harm done to society; and finally Co-evolutionary (very strong sustainability), where the company gives as much as it receives and understands the world as a place of human coexistence.

It is worth observing that, from the specific and shareholder-focused view proposed by Friedman [4] to the expanded view of stakeholders by Freeman and Reed [9], more grouping attempts were made by Carroll [3], Lantos [7], Garriga and Melé [5], and Landrum and Ohsowski [6], revealing a field that is becoming more complex and tends to open into multiple "angles", which these authors try to arrange and classify. Furthermore, if the field is now viewed from its specific relationship with different areas of management, it can be possible to observe the efforts made by authors who establish associations with finance, marketing, human resources, operations, and strategy, all of which generate the expansion and ratify the evolution of CSR.

On this line, Bosch-Badia et al. [11] pointed out that CSR was evolving from the focus on financial returns it had from 1990 to 2010, towards focusing on shared value [8]. This was corroborated by the Social Business Models (SBMs) proposal and the need to redefine the utility equation, value propositions, and value chain [12]. The argument is that SBMs will replace the traditional position of the shareholder with that of the stakeholder, steering capitalism towards current global concerns and adding concepts such as the co-creation of value [13].

The complexity of measuring the impact of CSR on financial results has also been acknowledged [14]. A link is established between Financial Performance (FP) and Corporate Social Performance (CSP) [15], justifying the positive but modest effect in the relationship between FP and CSR [16]. The conclusion is that the relationship between FP and CSR depends on the mediating effects of the company's intangible resources [17] and that better financial results can be achieved by stimulating innovation, human capital, reputation, culture, and trust [18]. Furthermore, by understanding organisational performance holistically

and not just from a financial perspective, CSR initiatives towards external stakeholders lead to positive financial results, mediated by the reputation of the organisation. [19].

An attempt has been made to model CSR as an investment that will help the company stand out and lead to benefits and higher earnings [20]. Long-term CSR activities are therefore found to increase value for the shareholder through a lower cash flow risk [21]. The argument is that different compliance standards associated with CSR ought be linked to risk exposure and costs in order to adopt effective CSR practices [22]. With regard to understanding CSR costs, it is suggested that their increase may be due to the company's desire to maximise its value and to the managers' interest in appearing benevolent in the eyes of the stakeholders [23], who, in turn, may be able to assess the companies' CSR performance using nonfinancial public information, shifting the conflict into scenarios other than a purely financial cost–benefit analysis [24].

In terms of CSR as part of the *strategy*, it is acknowledged that "CSR activities and practices are beneficial to businesses" [25], that effective strategies are specific and not generic [26,27], that those aimed at improving stakeholder relations and social welfare are more successful [28], and that they should be targeted at areas that demonstrate a convergence between economic and social objectives [8].

Studies have been carried out on CSR, the effective communication of its programmes [29] and transparency in communicating them, which would increase trust and reputation [30], the effect on consumers of CSR communication on social networks [31], and customers' credibility in organisations [32]. Research has also been performed on CSR and competitive contexts, studying its effect on customer purchase intention [33] and on the consumer's engagement with the brand [34]. In this field, literature findings show that CSR activities encourage the creation of better habits through marketing [35], with the introduction of new concepts such as co-creation as a collaborative process that goes beyond customer loyalty [36].

Attempts have been made to measure the impact of the size and composition of governing boards [37], the participation of women in such boards [38], which increases sensitivity to CSR [39] and to participative decision-making styles [40]. Findings show that including NGO executives on boards [41] and implementing gender diversity lead organisations to more CSR actions [42].

The literature proposes a revision of the business–society relationship, as, given the loss of the State's regulatory power, businesses are increasingly taking on a more political role [43]. This could call into question the need for profit and the existence of the corporation [44], and hence the various approaches on the theory of the firm, such as the agency theory [45], the stewardship theory [46], the team-based approach [47], inter alia [43]. CSR ratings are assumed to be higher for companies with fewer agency problems [48].

The efforts to guide, define, and study the practices of CSR-related actions from academia, together with the global guidelines that drive the development of responsible practices by organisations around the world, such as the Global Compact Principles, show a wide scope for the study of CSR. In addition to what has been mentioned, the literature delves into the different theories regarding each of the relevant business actors (entrepreneurs, managers, workers, lenders, insurers, academics, suppliers, etc.) and CSR, revealing key findings for organisational management and decision-makers.

Considering all of the above, this paper adds a bibliometric and scientometric analysis in order to understand how scientific interest in CSR has evolved. The goal is to be able to evaluate the possibility that CSR can begin to be considered as an area of specialisation within management or as a basic alternative to understand how management and business should currently be carried out. The authors believe that either path would leave room for an academic debate that could propose new ways of relating business and society.

## 2. Materials and Methods

The literature review shown in the introduction is complemented by a bibliometric and scientometric analysis. A scientometric analysis is carried out by applying bibliometric techniques to science [49]. It is the science that focuses on the quantitative study of scientific production, creating indicators that make analysis possible. The process focuses on scientific documents as empirical units of analysis. Among the main indicators used in scientometrics are the analysis of citations, which generate impact indicators, as well as the possibility of defining a set of parameters in order to measure the impact of journals and/or institutions, and the understanding of scientific citation theories, dividing them into normative and constructivist. A mapping of science is used to show the relationship between elements or aspects of science and scientologists, and indicators are developed to be used in policy and management contexts [50]. Given the growing interest of science and technology studies in the globalisation of knowledge production and the location of these activities in specific locations [51,52], the work was carried using information from the Web of Science (WoS) Science Citation Index (SCI-E) with "between 1975 and 2020" as an indicator, which covers most of the important international journals in the area of pure, applied, and medical sciences, and the Social Science Citation Index (SSCI), which covers the area of social sciences [49].

The structural aspects of the scientific community make it possible to deal with cases of associations by means of an analysis of co-authorships in publications and the degree of impact these associations have on institutions. Additionally, these aspects highlight the extent of cooperation between countries, institutions, authors, and shared references (co-referencing), that make it possible to establish scientific networks or to identify their belonging to a scientific discipline by analysing keywords that are shared in different research (co-words).

This establishes greater closeness by taking into consideration search vectors based on keywords, logical conjunction connectors and closeness restrictions [53] in the indexed papers. For this purpose, the key concept "Social Responsibility" was analysed in all languages but restricted to papers published in the "Business and Economics" category. The search yielded 8728 papers.

This research uses the following bibliometric indicators in the analysis: papers and citations in the area under study, authors with the greatest impact, most productive authors, main journals, WoS categories, main institutions and countries, in addition to a bibliometric analysis with the concept of social responsibility and map of key concepts and their respective clusters based on frequency data. The data obtained were studied through graph theory applied to social network using the VOSviewer software version 1.6.15., a user-friendly programme that makes it possible to easily build and view bibliometric network maps [54]. The programme can be downloaded from https://www.vosviewer.com/ (accessed on 27 April 2021).

The WoS database search, updated as of 27 April 2021, follows: (TS = ("Social Responsibility") and SU = Business and Economics) AND TYPES OF DOCUMENTS: (Paper) Index = SCI-EXPANDED, SSCI, A and HCI Time frame = 1975–2020. The analytical work was carried out between May and August 2021.

## 3. Results

The results presented in the introduction that were found while reviewing the literature demonstrate the efforts that have been made to describe and categorize CSR. Moreover, they show the academic link with the different areas of management. These results are complemented by the bibliometric and scientometric analysis presented in this section.

### 3.1. Papers and Citations in the Study Area

Considering that the aim is a broad understanding of the evolution of CSR, the search focused on papers dated between the years 1975 to 2020, dealing with the concept of "Social Responsibility". This makes it possible to incorporate the terms "Corporate", "Company",

and other nouns that can be associated with this concept into the analysis, in order to capture them all, without leaving any out; 8728 papers were identified, starting in 1975 with 12, with a linear growth of PAPER(YEAR) = 18,145(YEAR) − 36,055 with an $R^2$ = 57.7%. It can be stated that there was a linear growth during the first 27 years. In 2003, twice as many papers were published as in the previous year; from then on there was an exponential growth that kept on going until recent years, reaching the peak scientific production in 2020 with 1335 papers, 54.4% of which were published in the last five years. This demonstrates the increase in critical mass in this area of study (see Figure 1).

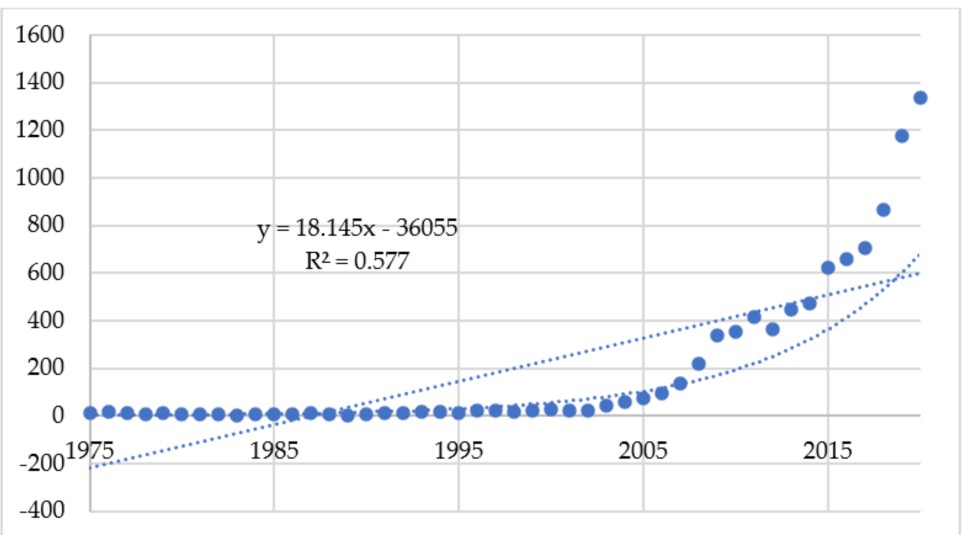

**Figure 1.** Growth of scientific production.

The total number of citations for Social Responsibility during this time reached 338,700, showing a similar performance as the publication of papers: it started in 1977 with two citations and increased linearly until the year 2000, when it began an exponential growth, reaching its peak in 2020 with 71,822 citations, accumulating 78.2% of citations in the last five years (see Figure 2).

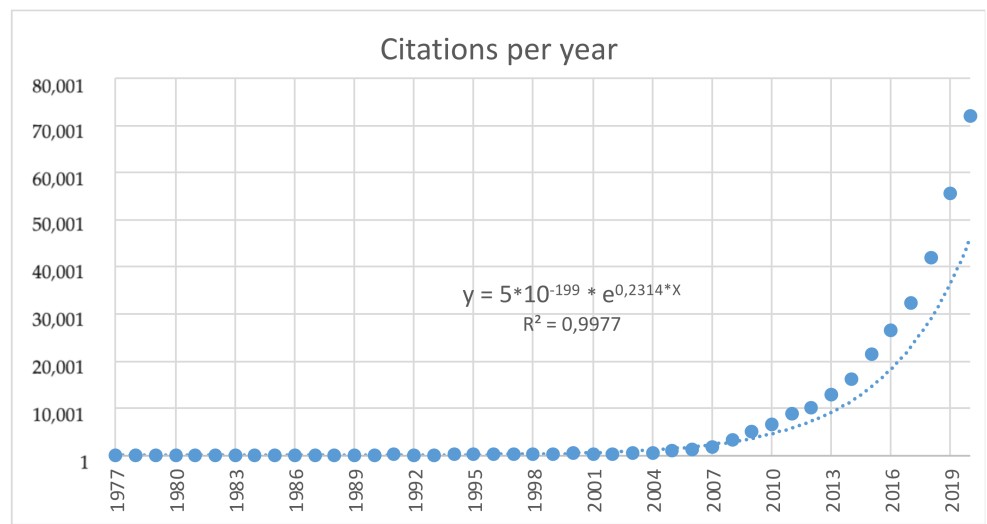

**Figure 2.** Total number of citations per year.

Table 1 shows the citation distribution of the papers in WoS. It emerges that 663 papers have no citations, 688 have 100 or more but less than 500 citations, and only 58 papers have more than 500 citations, with two of them cited more than 3000 times. It is believed

that the fact that only two papers have so many citations and only three reach more than 2000 citations reflect the broadness of the scientific field, which continues to grow.

**Table 1.** Citation general structure.

| Number of Citations | Number of Papers | % of Papers |
| --- | --- | --- |
| Over 3000 | 2 | 0.02% |
| more than 2000 less than 3000 | 3 | 0.03% |
| more than 1000 less than 2000 | 12 | 0.14% |
| more than 500 less than 1000 | 41 | 0.47% |
| more than 100 less than 500 | 688 | 7.88% |
| less than 100 | 7319 | 83.86% |
| 0 citations | 663 | 7.60% |
| Total citations | 8728 | 100.00% |

Source: Based on Web of Science data (2020) and compiled by the authors (2021).

To be highlighted among the 8728 papers is their Hirsch index or h-index [55], an index that generally favours long-standing authors who continuously publish papers with a long-lasting and above-average impact. Of the papers found, 228 exceed 228 citations, being the ones with the highest impact of the entire group of publications studied. Noteworthy is the paper by Donaldson and Preston [56] published by *Academy of Management Review* (Q1), which is cited 4220 times (1.25% of the total); in it the authors examined the three aspects of stakeholder theory, made criticisms, and added important contributions to the literature related to each aspect. The second most cited paper is by Michael Porter and Mark Kramer [8], with 3320 citations, representing 0.98% of the total citations. Published by *Harvard Business Review* (Q1), here the authors proposed a new way of looking at the relationship between business and society where business growth and social welfare is not a zero-sum game. This is achieved by identifying the social consequences of the firms' actions and discovering opportunities to benefit both them and society and by applying effective CSR initiatives.

In terms of total citations per paper, Table 2 lists the 16 papers considered the most influential and that have been cited at least 1000 times. Where it is possible to observe a wide range of concepts linked to CSR, some of which are transversal, such as Stakeholders Theory, Strategy and Society, Theory of the Firm, and others more specific, such as Consumers, Supply Chain Management, Financial Performance and Philanthropy, which is consistent with the results of the literature review.

**Table 2.** Most cited papers within the scientific productions.

| Ranking | Authors's Name | Year | Title | Journal | Total Citations |
| --- | --- | --- | --- | --- | --- |
| 1 | Donaldson, Preston | 1995 | The Stakeholder Theory of the Corporation—Concepts, Evidence, and Implications | *Academy of Management Review* | 4220 |
| 2 | Porter, Michael; Kramer, Mark | 2006 | Strategy and Society | *Harvard Business Review* | 3320 |
| 3 | McWilliams, Siegel | 2001 | Corporate Social Responsibility: A Theory of the Firm | *Academy of Management Review* | 2779 |
| 4 | Waddock, Graves | 1997 | The Corporate Social Performance–Financial Performance Link | *Strategic Management Journal* | 2684 |
| 5 | Sen, S; Bhattacharya, CB | 2001 | Does Doing Good Always Lead to Doing Better? Consumer Reactions to Corporate Social Responsibility | *Journal of Marketing Research* | 1974 |

**Table 2.** *Cont.*

| Ranking | Authors's Name | Year | Title | Journal | Total Citations |
|:---:|:---:|:---:|:---:|:---:|:---:|
| 6 | Campbell, John L. | 2007 | Why Would Corporations Behave in Socially Responsible Ways? An Institutional Theory of Corporate Social Responsibility | *Academy of Management Review* | 1802 |
| 7 | Brown, TJ; Dacin, PA. | 1997 | The Company and The Product: Corporate Associations and Consumer Product Responses | *Journal of Marketing Research* | 1774 |
| 8 | McWilliams, Siegel | 2000 | Corporate Social Responsibility and Financial Performance: Correlation or Misspecification? | *Strategic Management Journal* | 1498 |
| 9 | Carter, Craig R.; Rogers, Dale S. | 2008 | A Framework of Sustainable Supply Chain Management: Moving Toward New Theory | *International Journal of Physical Distribution and Logistics Management* | 1404 |
| 10 | Dahlsrud, Alexander | 2008 | How Corporate Social Responsibility Is Defined: An Analysis of 37 Definitions | *Corporate Social Responsibility and Environmental Management* | 1372 |
| 11 | Aguilera, Ruth V.; Rupp, Deborah E.; Williams, Cynthia A.; Ganapathi, Jyoti | 2007 | Putting the S Back in Corporate Social Responsibility: A Multilevel Theory of Social Change in Organizations | *Academy of Management Review* | 1342 |
| 12 | Luo, Xueming; Bhattacharya, C. B. | 2006 | Corporate Social Responsibility, Customer Satisfaction, and Market Value | *Journal of Marketing* | 1264 |
| 13 | Mcguire, JB; Sundgren, A; Schneeweis, T | 1988 | Corporate Social Responsibility and Firm Financial Performance | *Academy of Management Journal* | 1230 |
| 14 | Klassen, RD; Mclaughlin, CP | 1996 | The Impact of Environmental Management on Firm Performance | *Management Science* | 1192 |
| 15 | Porter, ME; Kramer, MR | 2002 | The Competitive Advantage of Corporate Philanthropy | *Harvard Business Review* | 1177 |
| 16 | Christmann, P | 2000 | Effects of Best Practices of Environmental Management on Cost Advantage: The Role of Complementary Assets | *Academy of Management Journal* | 1029 |

Source: Based on Web of Science data (2020) and compiled by the authors (2021).

### 3.2. Main Authors

In the 8728 papers, 14,196 authors are acknowledged as sole or co-authored researchers. The 10 most influential authors account for 17% of the citations. From Table 3 it can established that Bhattacharya from the University of Pittsburgh is the most influential author with 18 published papers on social responsibility and 8077 citations, representing 2.4% of the total, and based on the h-index with 9 papers among the 227 most influential. His most cited paper, with 1394 citations, links CSR to consumer satisfaction and market value [57]. Sankar Sen of the City University of New York is the second most influential author, whose 22 papers have been cited 6458 times, which places 7 of his publications among the 227 most influential papers in the period. Table 3 shows the most influential authors on Social Responsibility.

When analysing the most influential authors with VOSviewer, it can be seen that Bhattacharya relates *CSR* to Consumers, Stakeholders, Risks, Employees, Performance, Customer Satisfaction, Market Value, Marketing, Incentives, and Recruitment, while Sankar relates it to Leadership, Consumers, Stakeholders, Competitive Position, Business Returns,

and Consumer Ethics, while McWilliams relates it to Financial Performance, Theory of the Firm, and Competitive Advantage. This shows how CSR tends to be incorporated into different areas of management.

**Table 3.** The most influential authors on Social Responsibility.

| R | Author's Name | Institution | TP-RS | TC-RS | % | HA | TP | TC | T227 |
|---|---|---|---|---|---|---|---|---|---|
| 1 | Bhattacharya, C.B. | University of Pittsburgh | 18 | 8077 | 2.40% | 27 | 42 | 11,307 | 9 |
| 2 | Sen, Sankar | University of Pittsburgh | 22 | 6458 | 1.90% | 21 | 44 | 9960 | 7 |
| 3 | McWilliams, Abagail | University of Illinois System | 2 | 4457 | 1.30% | 15 | 22 | 7401 | 2 |
| 4 | Siegel, D. | Arizona State University | 10 | 4487 | 1.30% | 47 | 152 | 10,792 | 7 |
| 5 | Donaldson, Thomas | University of Pennsylvania | 1 | 4240 | 1.30% | 17 | 41 | 6039 | 1 |
| 6 | Preston, Lee E. | University of Maryland College Park | 1 | 4223 | 1.20% | 13 | 81 | 5268 | 1 |
| 7 | Brammer, Stephen | Macquarie University | 19 | 3476 | 1.00% | 27 | 44 | 5379 | 6 |
| 8 | Kramer, Mark | North West University—South Africa | 1 | 3396 | 1.00% | 1 | 3 | 3396 | 1 |
| 9 | Porter, Michael E | Harvard University | 1 | 3396 | 1.00% | 51 | 127 | 35,687 | 1 |
| 10 | Jamali, Dima | University of Sharjah | 26 | 2976 | 0.90% | 31 | 92 | 4208 | 4 |

Abbreviations: R: author's ranking; TP-RS: author's total number of papers on social responsibility; TC-RS: author's total citations of papers on social responsibility; HA: author's h-index; TP: author's total number of papers; TC: total number of citations per author; T227: author's total number of papers that are among the 227 most influential papers ever published. Source: Based on Web of Science data (2020) and compiled by the authors (2021).

The contribution to the generation of knowledge, in relation to the search vector, is determined by the number of papers published. These are not always the most influential authors, but they are important in terms of their scientific productivity. Hence, Table 4 lists the ten authors who are most productive related to social responsibility, indicating the number of papers on the subject, the total number of citations, the average number of citations of the published papers, the percentage on the total number of papers published on the subject, the author's h-index, the total number of publications by the author in the WoS platform as of April 2021, and the total number of citations of the author calculated on his or her publications in the WoS platform as of April 2021.

Table 4 shows that out of the 10 most prolific authors on social responsibility, only Dima Jamali from the University of Sharjah, who appears in fifth place with 26 published papers, also appears as an influential author in tenth place in Table 3, with 4 of her papers among the 227 most influential publications.

When analysing the most productive authors using VOSviewer, it emerges that, in her research, the author Garcia-Sanchez links CSR to Communications, Organisational Design, Stakeholders, Innovation, CSR practices, and Family Businesses. As for the next most productive authors with affiliation to other universities, Moon, who is the fourth most productive, relates his CSR research to Education, Gender, Codes of Conduct, Governance Systems, and Multinationals. Jamali, as the fifth most productive author, relates it to Corporate Governance, Developing Countries, Stakeholders, Investor, and People Management. This also shows the range of different fields to which CSR is linked.

Figure 3 displays a graph of the co-authorship analysis on the concept of social responsibility. The papers were input into the VOSviewer software, which groups the authors into clusters as detailed in Table 5.

Each cluster represents a set of authors who have teamed together to produce some of the scientific papers. These 14 clusters are identified in the Figure 3 graph each with their own specific colour; the colour is then specified in Table 5 under the cluster number, followed by a list of all the authors, with the most influential author of each cluster

highlighted in bold and italics. For example, in Table 5, Cluster 1 contains 12 authors of which the most influential is David Waldman, who is highlighted in bold and italics. This group of authors appears in Figure 3 graph in red, making it possible to determine at a quick glance the importance of this co-authorship network in comparison to the 13 remaining clusters. It can be observed that none of the authors leading the clusters is in the list of most influential authors (Table 3), nor among the authors with the most cited scientific production (Table 2).

**Table 4.** The most productive authors.

| R | Author's Name | University | TP-RS | TC-RS | PC-RS | % Tt | H-A | TP-A | TC-A |
|---|---|---|---|---|---|---|---|---|---|
| 1 | García-Sanchez, IM | University of Salamanca | 56 | 2253 | 40.2 | 64% | 35 | 164 | 4432 |
| 2 | Martinez-Ferrero, J. | University of Salamanca | 29 | 568 | 19.6 | 33% | 15 | 43 | 473 |
| 3 | Gallego-Alvarez, I. | University of Salamanca | 28 | 927 | 33.1 | 32% | 22 | 67 | 1613 |
| 4 | Moon, J. | University of Nottingham | 28 | 2261 | 80.8 | 32% | 29 | 67 | 5017 |
| 5 | Jamali, Dima | University of Sharjah | 26 | 2904 | 111.7 | 30% | 30 | 92 | 4096 |
| 6 | Kim, Jiyoung | University of North Texas Denton | 25 | 1247 | 49.9 | 29% | 25 | 64 | 2465 |
| 7 | Perez, A. | Universidad de Cantabria | 24 | 805 | 33.5 | 27% | 16 | 46 | 912 |
| 8 | Kim, Y. | Myongji University | 23 | 1421 | 61.8 | 26% | 8 | 18 | 226 |
| 9 | Kolk, A. | University of Amsterdam | 23 | 1673 | 72.7 | 26% | 5 | 14 | 64 |
| 10 | Lindgreen, A. | University of Pretoria | 23 | 1215 | 52.8 | 26% | 31 | 145 | 3878 |

Abbreviations: R: author's ranking; TP-RS: total papers of the author in the search vectors; TC-RS: total citations of the author's papers in the search vectors; PC-RS: total citations of the author's papers in the search vectors; PC-RS: average number of citations per paper in the search vectors (336/56); %Tt: percentage of total papers in the search vectors; H-A: author's h-index; TP-A: author's total number of papers; TC-A: total number of citations per author. **Source:** Based on Web of Science data (2020) and compiled by the authors (2021).

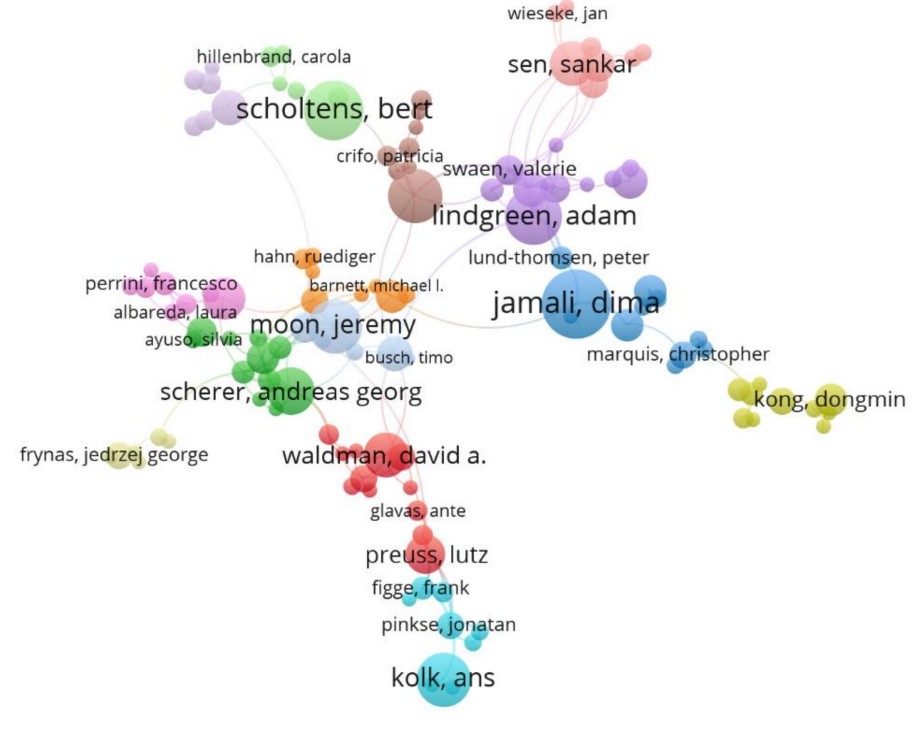

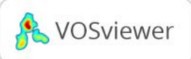

**Figure 3.** Co-authorship network for scientific production.

**Table 5.** Clusters on co-authorship for scientific production.

| Cluster 1 | Cluster 2 | Cluster 3 | Cluster 4 | Cluster 5 |
|---|---|---|---|---|
| **Red** | **Green** | **Blue** | **Yellow** | **Purple** |
| Aguinis, Herman | Arenas, Daniel | Amran, Azlan | Cheung, Yan-Leung | Adegbite, Emmanuel |
| Barketmeyer, Ralf | Ayuso, Silvia | Jain, Tanusree | Crilly, Donal | Amaeshi, Kenneth |
| Giuliani, Elisa | Baumann-Pauly, Dorothee | *Jamali, Dima* | *Kong, Dongmin* | De Roeck, Kenneth |
| Glavas, Ante | Baur, Dorothea | Liang, Hao | Ni, Na | Idemudia, Uwafiokun |
| Maak, Thomas | De Bakker, Frank G.A. | Lund-Thomsen, Peter | Peng, Mike W. | Janseen, Catherine |
| Miska, Christof | Palazzo, Guido | Marquis, Christopher | Qian, Cuili | *Lindgreen, Adam* |
| Pless, Nicola M. | Rasche, Andreas | Nejati, Mehran | Tan, Weiqiang | Maon, Francois |
| Preuss, Lutz | *Scherer, Andreas Georg* | Quazi, Ali | Tsang, Albert | Mzembe, Andrew N. |
| Siegel, Donald S. | Schneider, Anselm | Renneboog, Luc | Wang, Heli | Swaen, Valerie |
| Stah, Guenter K. | Spence, Laura J. | Toffel, Michael W. | Yu, Yangxin | Vanhamme, Joelle. |
| Voegtlink, Christian | Wickert, Christopher | Yin, Juelin | | |
| *Waldman, David A.* | | | | |
| **Cluster 6** | **Cluster 7** | **Cluster 8** | **Cluster 9** | **Cluster 10** |
| **Light blue** | **Orange** | **Brown** | **Purple** | **Pink** |
| Comyns, Breeda | Barnett, Michael L. | Crifo, Patricia | Albareda, Laura | Bhattacharya, C.B. |
| Figge, Frank | Crane, Andrew | Delmas, Magali A. | Castello, Itziar | Du, Shuili |
| Fransen, Luc | Gold, Stefan | Durand, Rodolphe | Lozano, Josep M. | Edinger-Schons, Laura |
| Hahn, Tobias | Hahn, Ruediger | *Gond, Jean-Pascal* | Misani, Nicola | Korschun, Daniel |
| Hansen, Eric | Henriques, Irene | Hawn, Olga | *Morsing, Mette* | Luo, Xueming |
| *Kolk, Ans* | *Husted, Bryan W.* | Ioannou, Ioannis | Perrini, Francesco | *Sen, Sankar* |
| Muller, Alan | Montiel, Ivan | Lyon, Thomas P. | Russo, Angeloantonio | Wieseke, Jan |
| Panwar, Rajat | Seuring, Stefan | Serafeim, Georg | Tencati, Antonio | Zheng, Qinqin |
| Pinkse, Jonatan | | | | |
| **Cluster 11** | **Cluster 12** | **Cluster 13** | **Cluster 14** | |
| **Opaque green** | **Opaque blue** | **Pale yellow** | **Pale purple** | |
| Dam, Lammertjan | Bondy, Krista | Brown, Jill A. | *Brammer, Stephen* | |
| Hillenbrand, Carola | Brusch, Timo | *Frynas, Jedrzej Georg* | Grosvold, Johanne | |
| Money, Kevin | Matten, Dirk | Khan, Zaheer | Habisch, Andre | |
| Oikonomou, Ioannis | *Moon, Jeremy* | Mellahi, Kamel | Millington, Andrew | |
| Hansen, Eric | Orlitzky, Marc | Park, Byung Il | Touboulic, Anne | |
| Pavelin, Stephen | Whelan, Glen | Soundararajan, Vivek | Walker, Helen | |
| *Scholtens, Bert* | | Brown, Jill A. | Brammer, Stephen | |

Source: Data from Web of Science (2020) and processed with VOSviewer software (2021).

The graph in Figure 4 displays the citations among the authors who have at least 5 published papers, hence restricted to 370 authors distributed in 10 clusters. The size of the circumference assigned to each author depends on the number of citations of their work. Hence, in Cluster 1 the author with the most citations is Adam Lindgreen (red), in Cluster 2 is Andrea Pérez (green), in Cluster 3 is Isabel Gallego-Álvarez (blue), in Cluster 4 is Bert Scholtens (yellow), among others.

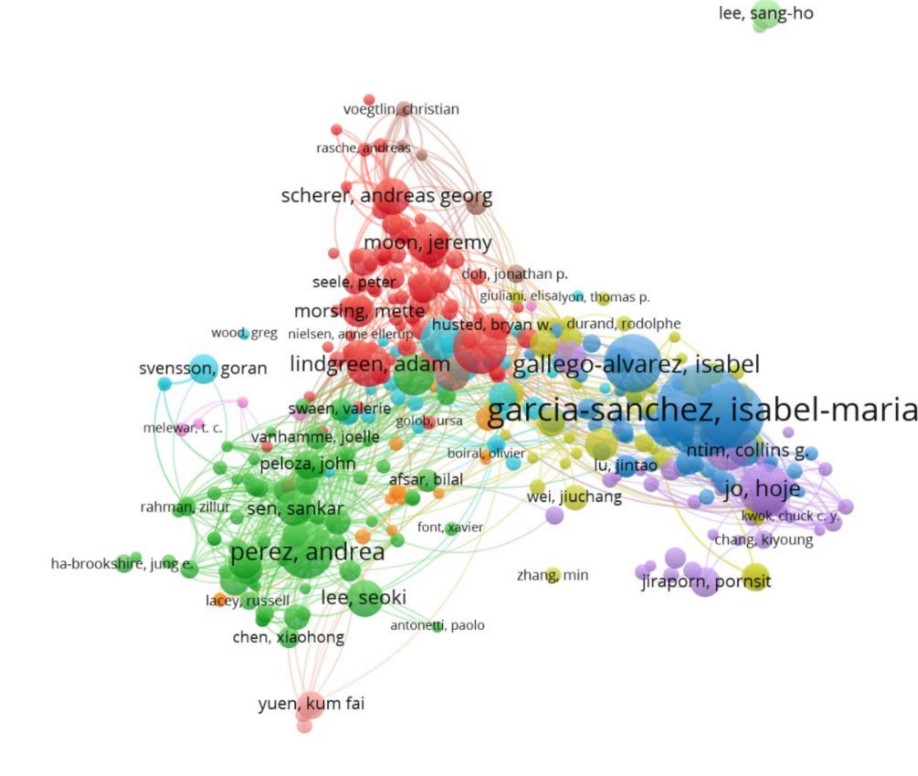

**Figure 4.** Network of joint bibliography for most cited scientific production.

*3.3. Main Journals*

The papers under study have been published in 556 journals, 10 have published 3650 papers concentrating 41.8% of all publications on the subject, with an average of 38.92 citations per paper, a total of 339,696 citations as a whole, and an h-index of 228. Table 6 details the 10 journals with the highest number of published papers.

**Table 6.** The highest scientific production.

| R | Sources (Journals) | NP | Tt % | PC-RS | H-RS | TC-RS | FI 5Y | Q |
|---|---|---|---|---|---|---|---|---|
| 1 | *Journal of Business Ethics* | 1758 | 20.14% | 52.44 | 145 | 92,188 | 5453 | Q2 |
| 2 | *Corporate Social Responsibility and Environmental Management* | 587 | 6.73% | 23.14 | 52 | 13,583 | 5485 | Q1 |
| 3 | *Business Strategy and the Environment* | 270 | 3.09% | 26.1 | 44 | 7047 | 6221 | Q1 |
| 4 | *Journal of Business Research* | 237 | 2.72% | 37.5 | 50 | 8888 | 5484 | Q1 |
| 5 | *Business Society* | 186 | 2.13% | 24.02 | 36 | 4468 | 4652 | Q2 |
| 6 | *Business Ethics: A European Review* | 178 | 2.04% | 24.12 | 32 | 4294 | 3423 | Q2 |
| 7 | *Public Relations Review* | 133 | 1.52% | 22.03 | 28 | 2930 | 2232 | Q3 |
| 8 | *Management Decision* | 126 | 1.44% | 25.21 | 33 | 3176 | 2886 | Q2 |
| 9 | *Amfiteatru Economic* | 89 | 1.02% | 8.85 | 17 | 788 | 1036 | Q4 |
| 10 | *Sustainability Accounting Management and Policy Journal* | 86 | 0.99% | 9.26 | 16 | 796 | 2056 | Q2 |
| | Summary | 3650 | 41.82% | 37.85 | 153 | 138,158 | 3893 | |

Abbreviations: R: ranking; NP: total number of papers only with social responsibility in the journal; Tt %: percentage of papers out of the total number of papers on social responsibility; PC-RS: average number of citations per paper in the search vectors; H-RS: h-index only with the search vectors; TC-SQ: total citations only with the search vectors; FI Y5: impact factor of the journal in the last 5 years; Q: quartile in the category. Source: Based on Web of Science data (2020) and compiled by the authors (2021).

In the analysis of Table 6, the *Journal of Business Ethics* stands out as having the largest number of papers (1758) and also as being the most influential, with the highest number of citations out of the total number of papers, i.e., with 92,188 out of a total of 339,696. The journal also stands out for the highest h-index with 145 and the highest citation average (52.44%). The highest impact factor of the past 5 years with 6221 goes to the journal *Business Strategy and the Environment*, with the impact factor being a measure of the quality of these journals.

From the point of view of this research, it is interesting to observe how journals from different areas of management and economics are willing to receive CSR papers. Even if it cannot be firmly stated, it is reasonable to think that, for this to happen, CSR must be associated with the areas of interest of each of the journals.

### 3.4. WoS Categories

The papers under study have been published in journals belonging to 10 WoS categories, with some of them published in different categories in parallel. The h-index of this category set reaches 43 with 5215 citations, 19.46 citations per paper, and referenced 1605 times, as detailed in Table 7. The table shows that the main contribution is from the Business category (65.23%) with the highest h-index reaching 205, the highest number of citations (255,532), and number of references by other papers (72,179). Additionally worth highlighting is the Ethics category with 2001 papers and 49.86 citations per paper. This could mean that, despite the fact that there are more publications in the Business and Management areas, the ethical perspective of CSR could be growing, as can also be observed in Table 6, where the *Journal of Business Ethics* has the highest number of papers published on Social Responsibility.

**Table 7.** Scientific production in WoS.

| R | Web of Science Categories | NP | % Tt | h-RS | PC-RS | TC-RS | AC |
|---|---|---|---|---|---|---|---|
| 1 | Business | 5693 | 65.23% | 205 | 44.89 | 255,532 | 72,179 |
| 2 | Management | 3751 | 42.98% | 161 | 38.93 | 146,016 | 56,995 |
| 3 | Ethics | 2001 | 22.93% | 146 | 49.86 | 99,767 | 37,600 |
| 4 | Economics | 1172 | 13.43% | 72 | 22.21 | 26,028 | 16,906 |
| 5 | Environmental Studies | 1155 | 13.24% | 77 | 24.41 | 28,192 | 17,772 |
| 6 | Business Finance | 871 | 9.98% | 88 | 34.56 | 30,103 | 12,790 |
| 7 | Hospitality Leisure Sport Tourism | 222 | 2.54% | 38 | 22.56 | 5008 | 3298 |
| 8 | Communication | 218 | 2.49% | 37 | 22.76 | 4961 | 3615 |
| 9 | Psychology Applied | 180 | 2.06% | 47 | 49.33 | 8879 | 6821 |
| 10 | Operations Research Management Science | 152 | 1.74% | 38 | 54.34 | 8259 | 6729 |
| | Summary | 15,415 | 99.44% | 228 | 39.09 | 339,227 | 92,107 |

Abbreviations: R: ranking; NP: total number of papers only on social responsibility in the journal; % Tt: percentage of papers out of the total number of papers on the search vectors; PC-RS: average number of citations per paper in the search vectors; h-RS: h-index only with social responsibility; TC-RS: total citations only in social responsibility; AC: number of papers with citations. Source: Based on Web of Science data (2020) and compiled by the authors (2021).

As with the main Web of Science journals where CSR papers are published, it is also interesting to note that the top 10 categories that receive these papers are rather varied, ranging from broad categories such as Business, Management, and *Economy* to specific topics such as Ethics, Environmental Studies, Finance, Applied Psychology, Communications, and Operations Research. This shows that CSR is related to the different management areas.

### 3.5. Institutions

Table 8 shows the 10 main institutions the researchers are affiliated with. They account for 8.6% of the total, and together they hold an h-index of 99 with 57.3 citations on average

and a total of 41,196 citations as a whole and in parallel in the same institutions. This is because some papers include more than one institution, with a total of 4058 having at least one paper published and with only three institutions exceeding 1% of the total number of publications. This low concentration on the subject indicates the topic is widely dispersed and is studied at different research centres.

**Table 8.** Institutions associated with scientific production, according to the author's affiliation.

| R | Organisations | Country | NP | % Tt | h-RS | PC-RS | TC-RS | AC |
|---|---|---|---|---|---|---|---|---|
| 1 | Copenhagen Business Sch. | Denmark | 97 | 1.11% | 34 | 53.89 | 5227 | 4378 |
| 2 | University of Nottingham | UK | 90 | 1.03% | 42 | 102.7 | 9243 | 7466 |
| 3 | York University | Canada | 89 | 1.02% | 35 | 50.29 | 4476 | 3991 |
| 4 | Penn State University | USA | 87 | 0.99% | 29 | 53.9 | 4689 | 4229 |
| 5 | University of Salamanca | Spain | 80 | 0.92% | 25 | 33.09 | 2647 | 1846 |
| 6 | Monash University | Australia | 66 | 0.76% | 25 | 34.27 | 2262 | 2121 |
| 7 | University of Michigan | USA | 64 | 0.73% | 30 | 59.81 | 3828 | 3358 |
| 8 | University of Bath | UK | 63 | 0.72% | 31 | 75.9 | 4782 | 4139 |
| 9 | University of Amsterdam | Holland | 60 | 0.69% | 28 | 54.67 | 3280 | 2935 |
| 10 | University of Manchester | UK | 55 | 0.63% | 23 | 45.84 | 2521 | 2353 |
| | Summary | | 751 | 8.6% | 99 | 57.3 | 41,196 | 23,653 |

Abbreviations: R: ranking; NP: total number of papers on social responsibility only; % Tt: percentage of papers out of total number of papers on social responsibility; h-RS: h-index with search vectors only; PC-RS: average number of citations per paper for the search vectors; TC-RS: total citations only with the search vectors; AC: number of papers with citations. Source: Based on Web of Science data (2020) and compiled by the authors (2021).

The bibliometric analysis of citations related to institutions with a minimum of 25 published papers shows 59 institutions and 6 clusters, which are detailed in Table 9. Figure 5 graph shows the correlation between the various institutions with different colours for each of the 6 clusters.

**Table 9.** Citations between institutions.

| Cluster 1 | Cluster 2 | Cluster 3 | Cluster 4 | Cluster 5 | Cluster 6 |
|---|---|---|---|---|---|
| **Red** | **Green** | **Blue** | **Yellow** | **Purple** | **Light Blue** |
| Bocconi Univ | Aalto Univ | Harvard Univ | City Univ Hong Kong | Amer Univ Beirut | Univ Cantabria |
| Concordia Univ | Arizona State Univ | Korea Univ | Hong Kong Polytech Univ | Deakin Univ | Univ Groningen |
| Erasmus Univ | Cardiff Univ | Monsah Univ | Huazhong Univ Sci and Tech | Macquarie Univ | Univ Salamanca |
| Florida State Univ | City Univ London | Santa Clara Univ | Nanyang Technol Univ | Univ Manchester | Univ Valencia |
| Temple Univ | Copenhagen Business Sch. | Tilburg Univ | Rmit Univ | Univ Southampton | Univ Zaragoza |
| Univ Technol Sydney | Univ Amsterdam | Univ Alberta | Shangai Jiao Tong Univ | York Univ | |
| Griffith Univ | Univ Hull | Univ Reading | Univ Bath | | |
| Indiana Univ | Univ London | | | | |
| Insead | Univ Nottingham | | | | |
| Northeastern Univ | Univ St Gallen | | | | |

**Table 9.** *Cont.*

| Cluster 1 | Cluster 2 | Cluster 3 | Cluster 4 | Cluster 5 | Cluster 6 |
|---|---|---|---|---|---|
| Red | Green | Blue | Yellow | Purple | Light Blue |
| Penn State Univ | Univ Warwick | | | | |
| Univ Calgary | Univ Zurich | | | | |
| Univ Cent Florida | | | | | |
| Univ Illinois | | | | | |
| Univ Leeds | | | | | |
| Univ Michigan | | | | | |
| Univ Minnesota | | | | | |
| Univ N. Carolina | | | | | |
| Univ Tennessee | | | | | |
| Univ Washington | | | | | |
| Univ Western Ontario | | | | | |
| Univ Wisconsin | | | | | |

Source: Data from Web of Science (2020) carried out with VOSviewer software.

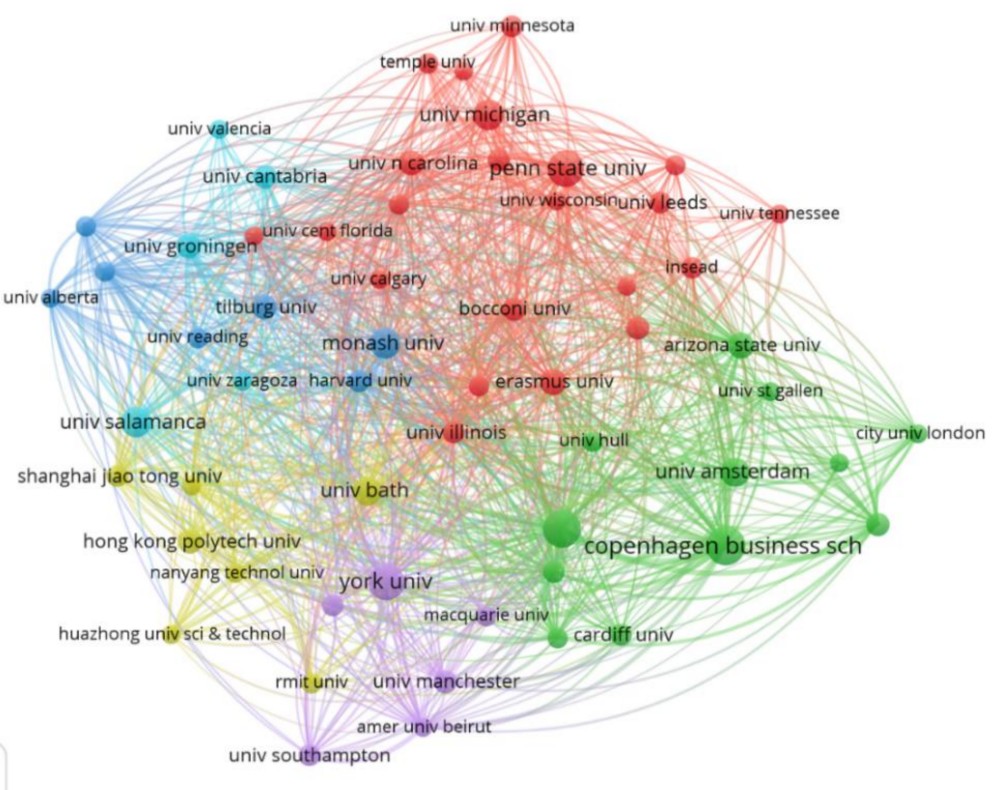

**Figure 5.** Network of the most cited institutions.

Figure 5 graph shows the six clusters. Penn State University prevails in Cluster 1 with 3497 citations and a relation with 479 other institutions; Cluster 2 is dominated by Copenhagen Business School with 4760 citations and 802 connections to other institutions; Cluster 3 by Monash University with 1902 citations and 279 connections with other institutions; Cluster 4 by Bath University with 4173 citations and a relation with 549 institutions; Cluster 5 by York University with 3506 citations and related to 575 institutions; finally, the

University of Salamanca prevails in Cluster 6 with 1918 citations and is related to 357 institutions. This graph shows the large number of relationships between the most productive institutions dealing with CSR; it confirms that the production of papers is concentrated in developed countries, and that, alongside the growth, the networks between research centres are growing significantly. The authors believe that in this case any conclusions in relation to the topic under study cannot be reached, as it is limited by VOSviewer for a deeper analysis.

*3.6. Countries*

This knowledge has been developed in a highly concentrated geographical area: 99.8% of the papers belong to 11 of the 113 countries that have produced at least one paper related to the subject under study. Countries that have published at least 300 papers related to social responsibility are listed in Table 10. The combined h-index of these countries is 219 and the average citation rate 42.07 per paper, with a total of 291,841 citations and 84,082 citations of these papers by other countries.

**Table 10.** Countries associated with scientific production, according to the author's affiliation.

| R | Countries/Regions | NP | % Tt | h-RS | PC-RS | TC-RS | AC |
|---|---|---|---|---|---|---|---|
| 1 | United States | 2690 | 30.82% | 180 | 58.72 | 157,968 | 59,243 |
| 2 | England | 1151 | 13.19% | 108 | 44.62 | 51,360 | 28,521 |
| 3 | China | 888 | 10.18% | 66 | 22.25 | 19,761 | 11,780 |
| 4 | Spain | 727 | 8.33% | 75 | 30.72 | 22,336 | 13,631 |
| 5 | Australia | 674 | 7.72% | 76 | 37.26 | 25,112 | 17,744 |
| 6 | Canada | 626 | 7.17% | 90 | 51.37 | 32,159 | 20,901 |
| 7 | France | 490 | 5.62% | 59 | 28.99 | 14,203 | 10,773 |
| 8 | Germany | 421 | 4.82% | 62 | 35.75 | 15,050 | 11,310 |
| 9 | Italy | 395 | 4.53% | 57 | 28.79 | 11,372 | 8526 |
| 10 | Netherlands | 349 | 3.99% | 75 | 53.49 | 18,668 | 13,498 |
| 11 | South Korea | 303 | 3.47% | 44 | 29.27 | 8868 | 6547 |
| | Total data | 8714 | 99.80% | 219 | 42.07 | 291,841 | 84,082 |

Abbreviations: R: ranking; NP: total number of papers related to social responsibility; % Tt: percentage of papers of the search vectors over the total number of papers of the same search vectors; h-RS: h-index only in social responsibility; PC-RS: average number of citations per paper on the search vectors; TC-RS: total citations only with the search vectors; AC: number of papers with citations. Source: Based on Web of Science data (2020) and compiled by the authors (2021).

Table 10 proves that the most productive country is the USA, generating 2690 papers related to social responsibility and 157,968 citations, making it the most influential and the area with the highest h-index (180); 59,243 of their papers have been cited and they have the highest average number of citations per paper, reaching 58.72.

The graph in Figure 5 corresponds to the co-authorship between countries, and it shows that at least 42 out of 113 countries have 20 or more papers co-authored. We found 6 clusters shown with different colours in Figure 5 that are detailed in Table 11.

In the graph shown in Figure 6, the most predominant circumference is the USA, which belongs to Cluster 2, while England prevails in Cluster 1, and Spain, which belongs to Cluster 3, appears in third place. It should also be mentioned that the size of the circumference is proportional to the degree of co-authorships by the authors belonging to these countries.

**Table 11.** Clusters of co-authorship between countries.

|  | Cluster 2 | Cluster 3 | Cluster 4 | Cluster 5 | Cluster 6 |
|---|---|---|---|---|---|
| **Red** | **Green** | **Blue** | **Yellow** | **Purple** | **Light Blue** |
| Austria | Australia | Brazil | Finland | Belgium | People's R China |
| Czech Republic | Japan | Canada | Italy | Denmark | Taiwan |
| England | New Zealand | Lebanon | Netherlands | France | |
| Germany | Singapore | Mexico | Norway | Pakistan | |
| Greece | South Africa | Portugal | Scotland | Wales | |
| India | South Korea | Spain | Sweden | | |
| Ireland | Thailand | UAE | | | |
| Lithuania | USA | | | | |
| Malaysia | | | | | |
| Poland | | | | | |
| Romania | | | | | |
| Slovenia | | | | | |
| Turkey | | | | | |

Source: Compiled by the authors based on VOSviewer.

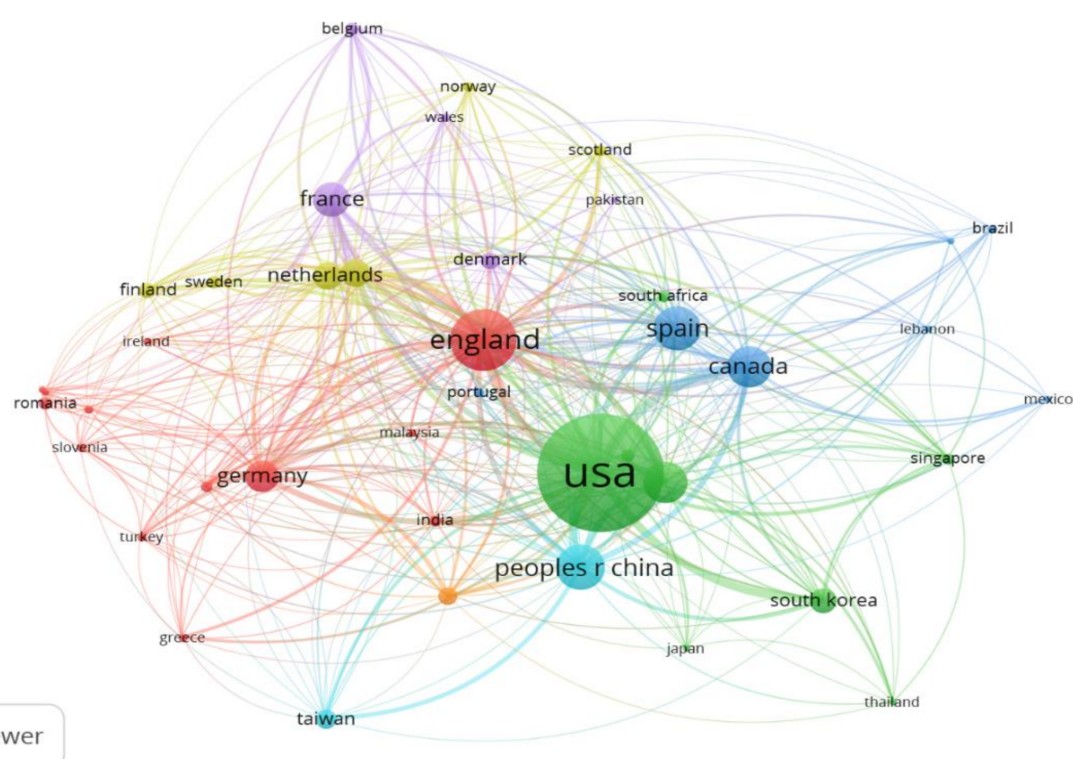

**Figure 6.** Co-authorship network between countries.

*3.7. Bibliometric Analysis of Keywords*

The WoS analysis shows 11,092 keywords, 123 of which appear 20 or more times and are used concurrently, as shown in Figure 6; this forms 8 clusters, arranged as detailed in Table 12.

**Table 12.** Co-occurrence clusters in the use of Keywords Plus.

| | |
|---|---|
| Cluster 1 22 items (red) | Advertising–Brand equity–Case study–Cause-related marketing–Commitment–Communication–Consumer behaviour–Consumer behaviour–Corporate identify–Corporate image–Corporate reputation–*Corporate social responsibility*–Customer loyalty–Customer satisfaction–Loyalty–Marketing–Public relations–Qualitative research–Reputation–Satisfaction–Social media–Trust |
| Cluster 2 18 items (green) | Banks–Content analysis–Corruption–CSR disclosure–CSR reporting–Disclosure–Earnings management–Environment–Environmental disclosure–Family firms–Global reporting initiative–Integrated reporting–Legitimacy theory–Regulation–Reporting–Socially responsible investing–Sustainability reporting–Voluntary disclosure |
| Cluster 3 16 items (blue) | Corporate citizenship–Corporate financial performance–Corporate responsibility–Corporate social performance–Diversity–Employees–Environmental sustainability–Ethical leadership–Firm performance–Gender–Human resource management–Leadership–Organizational identification–Resource-based view–Scale development–Stakeholder management |
| Cluster 4 16 items (yellow) | Agency theory–Business and society–China–Community–Corporate philanthropy–Corporate social responsibility (CSR)–CSR communication–Developing countries–Emerging markets–India–Institutional theory–Mining–Multinational enterprises–Philanthropy–Stakeholder theory–Stakeholders |
| Cluster 5 15 items (purple) | Business ethics–Competitive advantage–Competitiveness–CSR–Culture–Entrepreneurship–Ethics–Innovation–Management–Organizational culture–Performance–SMEs–Social capital–Social responsibility–Strategy |
| Cluster 6 14 items (light blue) | Assurance–Board of directors–Business strategy–Climate change–Corporate governance–Environmental performance–Environmental policy–Financial performance–Firm value–Gender diversity–Spain–Stakeholder engagement–Sustainable development |
| Cluster 7 13 items (orange) | Accountability–Environmental–Globalization–Governance–Human rights–Institutions–Legitimacy–Multinational corporations–Power–Responsibility–Stakeholder–Supply chain–Transparency |
| Cluster 8 9 items Pink | Corporate sustainability–Environmental management–Environmental responsibility–Event study–Risk management–Socially responsible investment–Supply chain management–Sustainability–Triple bottom line |

Source: Web of Science data (2020).

The graph in Figure 7 shows many interconnections between these concepts, and Table 10 groups them as clusters, recognising the emphases around which the papers under study are developed. The graph shows that the most used keyword is "Corporate Social Responsibility" corresponding to Cluster 1 with 2182 occurrences. "Sustainability" occupies the second place with 389 occurrences corresponding to Cluster 8. "Social Responsibility" with 326 occurrences corresponding to Cluster 5 occupies the third position. These three keywords have several interconnections with the majority of keywords.

The large number of keywords associated with the concept of social responsibility shows how cross-sectional the topic is and how it has been introduced into different scientific and disciplinary areas of science. The variety of words within the same cluster that separately seem so distant—for instance, in Cluster 3 with keywords such as Corporate Financial Performance, Diversity, Environmental Sustainability, Ethics, Leadership, Gender, and Human Resources—specifically shows how they are established links to very different study areas, which could support the idea that it can be considered an area of specialisation within the management of businesses.

A breakdown of the 10 most recurring keywords is presented in Table 13.

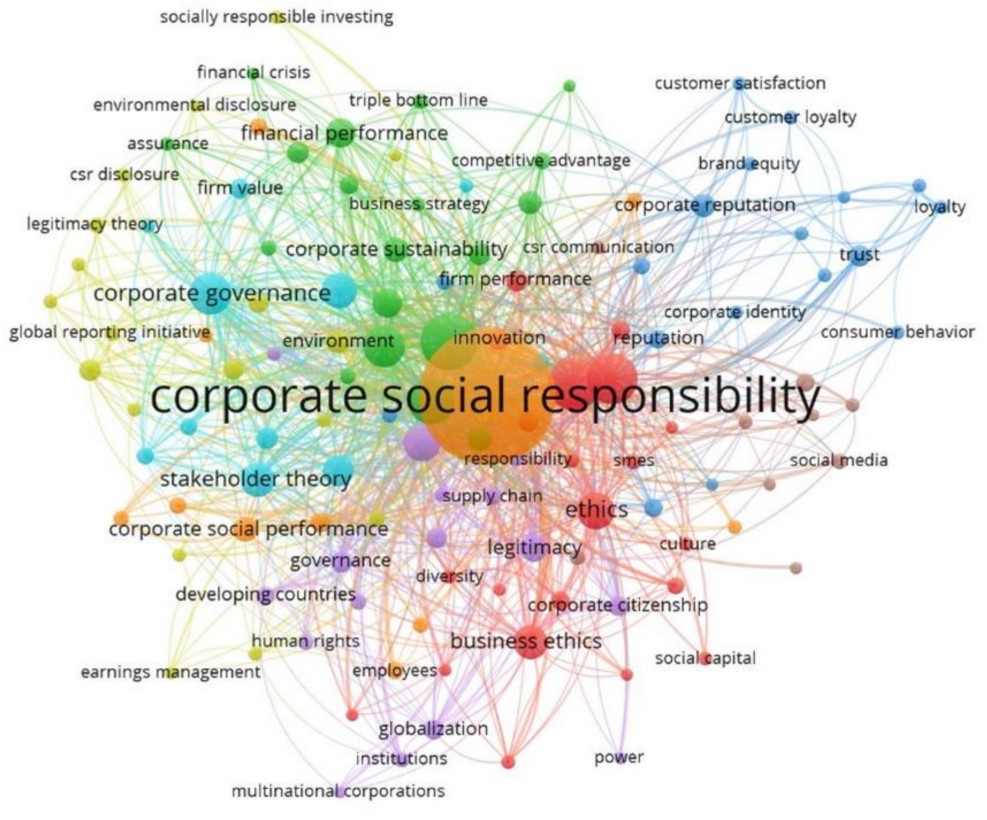

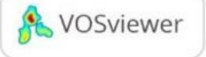

**Figure 7.** Bibliometric network of the research on social responsibility.

**Table 13.** Co-occurrence clusters in the use of Keywords Plus.

| # | Keyword | Occurrence |
|---|---|---|
| 1 | Corporate social responsibility | 2182 |
| 2 | Sustainability | 389 |
| 3 | Social responsibility | 326 |
| 4 | CSR | 319 |
| 5 | Corporate Social Responsibility (CSR) | 268 |
| 6 | Corporate governance | 208 |
| 7 | Stakeholders | 173 |
| 8 | Ethics | 172 |
| 9 | China | 150 |
| 10 | Business ethics | 148 |

Source: Web of Science data (2020).

## 4. Discussion

The combined results of the bibliographic and scientometric analysis confirm the complexity of the field and exponential growth of the interest in CSR. The Web of Science (WoS) shows growing production rates with 8728 papers published between 1975 and 2020 in the Business and Economics category for the "Social responsibility" concept, increasing from 12 publications in 1975 and a lineal growth up to 2002, to an exponential growth from 2003, reaching 1335 papers published in 2020, which demonstrates scholars' interest in this field. The literature review shows that from 1995 efforts are made to categorise CSR, to define concepts related to it, and evidenced that the subjects related to this field of study are

varied. CSR therefore appears to be linked to general management areas such as Marketing, Finance, Operations, and Human Resources and from there to more specific areas such as Consumers, Corporate Finance, Supply Chain, and Leadership.

Although the bibliometric and scientometric analyses are limited as they only provide a snapshot of the period when analysis was performed and do not show an evolution over time of the CSR keywords, they do illustrate the breadth of related topics (as can be seen in the co-occurrence clusters), the diversity among the same authors researching CSR topics by relating them to different areas of management, as shown in the analysis of the research by the most influential and most productive authors.

The authors believe that when the findings of these analyses are integrated with a thorough revision of the literature, there are sufficient reasons to believe in the importance that CSR is viewed as a specialisation area within organisational management, or as a new way to understand organisational management, whereby each business process ought to be socially responsible.

New approaches to CSR could help to start a debate about a business–society relationship that makes it possible to move away from "profit generation as the ultimate goal" [7], or move from a purely instrumental approach to a more integrative approach, integrating social demands and stakeholder interests [5]. The authors believe that any of the proposed new ways of understanding CSR could help to take companies from an "only compliance (weak sustainability) management style" to regenerative (strong sustainability) and co-evolutionary (very strong) levels [6].

This leads to a new roadmap for the development of CSR research, whereby each discipline or practice in business can now be understood as an object of analysis to be questioned in terms of the impact it has on each of the stakeholders, who could either be affected by its implementation or who could determine its development.

This approach makes it possible to move away from the complementary role that CSR has in management, whereby a company, after categorizing its business processes, can analyse how each one of them relates to the stakeholders and, from there, promote responsible management in the areas concerning the different stages of the process. The understanding of business as activities that should be systematically viewed as acts of responsibility linking the companies that run a business with a society that needs business, thus becomes a broad field of study that opens a new perspective whereby CSR to be incorporated as a central axis of business development, from which it might be possible to improve the relationship between the business goals and society goals.

With these premises, this paper identified analyses linked to the planning, strategy, design, and organisational structure that lay the foundations for business in a socially responsible context [12,37,38,41–43,58,59]; those that provide critical guidelines for the management of the various stakeholders [33,35,36,60–64]; those that seek to measure, control, and report on responsible management [6,14,17,42,65–67]; CSR and performance management research [18,20,21,23,25,48,68]; and, finally, those that highlight strategic communication and transparency in organisational management as drivers of social responsibility [29–31,69,70]. CSR is being linked to all areas of management and it seems to require a systematisation of the way it is understood and applied.

Considering the limitations of scientometric analyses, which do not allow for the conceptual structure of the field of study to be analysed, a future line of research could be an in-depth investigation on the development of the key concepts of CSR in each period under consideration. This would allow correlation of the key concepts of the research area, not only in a static way, but also by comparing different scenarios over time, which would make it possible to understand the state of the art and the evolution of the concepts revolving around CSR and, in this way, try to understand where it is heading.

In addition to this, for new research to be developed, it is important to organise the topics associated with the study of CSR in the areas where the growth in academic production and the diversification of relevant topics have made it difficult to understand the concepts associated with this field. Ultimately, redefining the most relevant concepts

would be of great help in the development of new studies linked to the field, as it would lead to a homogeneous collective understanding, thereby promoting the depth and wider scope of the research findings.

Future research processes ought to reinforce the idea of CSR as a specific and systemic area that demands management and organisational efforts, as a new way of understanding business. Alternatively, they ought to contrast both the findings from the literature review, this scientometric analysis, and others, which can be carried out by applying new tools available to the scientific world.

Finally, the authors consider that, although the results lead to the conclusion that CSR ought to be transformed into a professional field or considered as an area of specialisation within organisational management, the scientometric analysis did not provide sufficient evidence to determine from which scientific area this could be performed. This is one of the main reasons why this analysis should be broadened with a second paper.

Nonetheless, the authors believe that, in order to create new approaches that connect business and society, various social science disciplines such as business, sociology, and anthropology should work together. By integrating the work of this and other disciplines, it may be possible to identify and explore new variables for designing business models that are not currently taken being considered from the traditional management perspective. This would generate research spaces so that science can help think new ways of developing business by applying a CSR approach.

**Author Contributions:** Conceptualization, M.M.-P.; methodology, M.M.-P., H.M-F. and L.A.-C.; validation, M.M-P., L.A.-C. and F.M.-L.; formal analysis, M.M.-P., L.A.-C., F.M.-L. and H.M.-F.; investigation M.M.-P.; data analysis H.M.-F. and M.M.-P.; supervision, L.A.-C. and F.M.-L.; writing and editing the original draft, M.M.-P. and H.M-F.; review, M.M.-P., L.A.-C. and F.M.-L. All authors have read and agreed to the published version of the manuscript.

**Funding:** This research received no external funding.

**Institutional Review Board Statement:** Not applicable.

**Informed Consent Statement:** Not applicable.

**Data Availability Statement:** The data will be made available on request from the corresponding author.

**Conflicts of Interest:** The authors declare no conflict of interest.

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
