# Peer review of "Scientometric Analysis of Research on Corporate Social Responsibility"

_sustainability, doi:10.3390/su14042291_

Round 1

Reviewer 1 Report

The manuscript is a special review article. Although it includes data analysis and gives a conclusion, it differs from classic research papers in the field. Such papers are at increasing popularity, especially based on the analysis tools of the big publishing houses. Despite the limitations, I can recommend the publication of the manuscript, I have no change requests. Its approach is valuable, technical quality and scientific soundness are great. The content can be interesting to the readers and researchers dealing with CSR.

Based on a Turnitin check, the manuscript can be considered original.

Reviewer 2 Report

This paper summarizes the literature of CRS in business and the economy, which is of outstanding contribution and importance to this field. In addition, this research topic is also very novel.

Through the analysis and description of VOSviewer, we can understand the evolution of CRS and its relationship with different subjects, which echoes the research objectives.

 But I think the article will be more remarkable if it improves in the following aspects. My suggestions are as follows for reference:

  1. Some English grammar modifications, such as L14 an insight of (change to "into"), classified companies in (change to "into")
  2. Try to merge the paragraphs of the introduction to highlight the main idea of each paragraph and the logical relationship between paragraphs.
  3. The conclusion is too rough to explain the findings combined with research and analysis.
  4. It does not correspond to the research purpose. For example, it does not conform to the improvement of stakeholders and social welfare mentioned in the introduction.
  5. Although there is a conclusion that assessment can become a professional field, it is not specified from management and in kind.

The above suggestions are for reference, and we look forward to your supplementary instructions.

Reviewer 3 Report

The paper is a typical scientometric/bibliometric analysis paper. However, the authors performed this analysis very well and clearly presented this in the paper, including all the necessary details supporting the results ant the conclusions.

The main remarks:

1)  I would like to see the exact keywords that were used for the search. Saying that “the search focused on papers …dealing with the concept of "Social Responsibility”, and including the terms "Corporate", "Company" and other nouns” is not specific enough.

2) You state that 8,728 papers were identified. Usually, after initial search, screening of papers is made and a part of papers is eliminated from a further analysis, because they are not directly related to research topic. Have you made this?

3) I still prefer to see conclusions and not only discussion.

4) Could you please shortly review the latest trends in 2021-2022, because the analysis involves papers only up to 2020.

Minor comments:

WoS, but not Wos; Quoted or cited?

Round 2

Reviewer 2 Report

Thank you for your reply, and I look forward to your successful publication.
